# Treatment of Infection as a Core Strategy to Prevent Rifampicin-Resistant/Multidrug-Resistant Tuberculosis

**DOI:** 10.3390/pathogens12050728

**Published:** 2023-05-17

**Authors:** Anja Reuter, Jennifer Furin

**Affiliations:** 1The Sentinel Project on Pediatric Drug-Resistant Tuberculosis, Cape Town 7405, South Africa; 2Department of Global Health and Social Medicine, Harvard Medical School, Boston, MA 02115, USA

**Keywords:** tuberculosis, drug-resistance, prevention, infection, post-exposure management

## Abstract

An estimated 19 million people are infected with rifampicin-resistant/multidrug-resistant strains of tuberculosis worldwide. There is little done to prevent these individuals from becoming sick with RR/MDR-TB, a disease that is associated with high rates of morbidity, mortality, and suffering. There are multiple phase III trials currently being conducted to assess the effectiveness of treatment of infection (i.e., “preventive therapy”) for RR/MDR-TB, but their results are likely years away. In the meantime, there is sufficient evidence to support a more comprehensive management of people who have been exposed to RR/MDR-TB so that they can maintain their health. We present a patient scenario and share our experience in implementing a systematic post-exposure management program in South Africa with the goal of inspiring similar programs in other high-burden RR/MDR-TB settings.

## 1. Introduction

Each year, more than a half million people become newly sick with rifampicin-resistant/multidrug-resistant forms of tuberculosis (RR/MDR-TB) [1]. The past decade has witnessed major advances in the way that RR/MDR-TB is diagnosed and treated. More needs to be done, however, to prevent morbidity, mortality, and suffering from this complicated infectious disease [2]. The public health approach to RR/MDR-TB has primarily focused on treating people after they become sick or conceptualizing prevention in a narrow frame aimed at preventing the selection of additional drug resistance among people who are on treatment for TB [3]. Within this review, we seek to reframe RR/MDR-TB prevention from a family-centered, equity-driven perspective. We emphasize actions that should be implemented without delay to address this ongoing crisis of antimicrobial resistance.

## 2. One Family’s Experience

RR/MDR-TB can have a major impact on a family. In our work in a peri-urban township located outside of Cape Town, we encountered many households struggling with the disease. As an example, the Makhanda (name changed to protect privacy and permission given to share this family experience. This family experience was typical of what was encountered by many families we worked with in South Africa) family struggled to meet the growing household’s most basic needs. With only one working household member, money was tight, and there was often not enough to eat. When the oldest adult brother died in a motor vehicle accident, the family matriarch, Busisiwe, had to make room for his three children—all under the age of ten years—to come and live in the already crowded shack where she was living with her daughter, her son, and four of her other grandchildren. Over the coming months, Busisiwe attributed her weight loss to the meagre food supply and the stress, but it was harder to explain away her cough. When she finally presented to the health center and was told she had RR/MDR-TB, she was devastated at the loss of her income that the daily visits to the health center for observed therapy would cause and she was worried about the family members living with her. Her son and two of the grandchildren were coughing as well, and although the nurse told her to bring all the household members to clinic for evaluation, they only had taxi fare to bring the sickest one. When she used that money to bring her grandson to clinic, she was grief-stricken to learn he was also sick and wept as they took him away to the hospital. She worried endlessly about the other children but was told there was nothing else they could do. “Just bring them to the clinic if they are sick” the nurse instructed her, as Busisiwe left the health center once again with a heavy heart, worried that she had brought this misfortune on them all. 

## 3. Epidemiology of RR/MDR-TB Infection 

Infection with *Mycobacterium tuberculosis* means that a person has the bacilli living in the lungs but is not yet sick with TB disease. In this paper, we will refer to this state as RR/MDR *M.tb* infection. The precise data on the number of people who are infected with RR/MDR-TB are not possible to calculate since current tests for TB infection cannot differentiate between drug-susceptible and drug-resistant strains. This is because drug susceptibility testing can only be done when *M. tuberculosis* or its DNA has been isolated, a condition that usually defines TB disease. Modelling studies suggest that there are more than 19 million people worldwide who harbor mycobacterial strains that are resistant to rifampicin and isoniazid [4] (the definition of multidrug-resistant TB) in their lungs. Given that about 500,000 people become sick with RR/MDR-TB each year and that the global average household size is four people, this means more than a million people are exposed to RR/MDR-TB annually [5]. While not all these individuals are infected with RR/MDR-TB and not all those who are infected will become sick, these numbers suggest that the growing reservoir of people with RR/MDR *M.tb* infection far outstrips treatment capacity. That fact alone is reason for concerted action, but until recently, the only measure recommended to prevent RR/MDR-TB among those who were exposed was frequent clinical monitoring [6].

Much of this stems from the fact that there are not yet randomized controlled trials supporting the use of a specific medication for the treatment of RR/MDR *M.tb* infection (or what is more commonly called preventive therapy). Some of this also stems from the historical debate about the transmissibility of RR/MDR *M.tb* strains, with some studies suggesting that the fitness costs associated with some mutations leading to resistance could reduce the transmissibility of these forms of TB [7]. Several epidemiologic studies have shown that household members of persons who are newly diagnosed with RR/MDR-TB have a markedly elevated risk of TB infection and disease, perhaps owing to the delays faced in many places in diagnosing and treating RR/MDR-TB [8,9].

## 4. Optimizing RR/MDR-TB Prevention through Systematic Post-Exposure Management

Rapid diagnosis and prompt initiation of effective therapy are among the most effective means for preventing RR/MDR-TB since primary transmission is driving most of the burden of RR/MDR-TB in many settings in the world [10,11]. Thus, the widespread use of WHO-recommended rapid molecular tests coupled with access to newer drugs and shorter regimens that can effectively treat many types of RR/MDR-TB in as little as six months and with as few as four drugs [12] is key to decreasing the reservoir of people infected with RR/MDR-TB mycobacterial strains. Offering such therapy in decentralized and supportive settings with counseling and socioeconomic support is also essential in RR/MDR-TB prevention. More needs to be done to make the treatment of people with all forms of TB person-centered [13].

When someone is newly diagnosed with RR/MDR-TB, working with them to compassionately disclose this diagnosis to other household members is key. If done in a supportive manner—as is the case with HIV and other infectious diseases [14]—and by avoiding blaming concepts and terms (see Text Box 1), then the entire household can be engaged in the next stages of care for all members [15]. This shift from the traditional concept of contact tracing to one of post-exposure management may also help spur reluctant health care services into more active modes of support [16]. Contact tracing could also be expanded beyond the household and into entire communities with high rates of RR/MDR-TB using the same strategies we describe here. 

Box 1Key Components of Disclosure Counseling.Empathize with the person who was just diagnosed;Assess any physical or mental risks that might result if disclosure happens;Identify any individuals who might be able to support disclosure with other household members and share the news with that individual first;Practice the disclosure and review what is the best way to say things for the newly diagnosed person;Emphasize the concepts of “shared air” rather than contagion and blame;Offer to assist the newly diagnosed person with the disclosure if s/he would like;Stress that there are many actions that can be done to keep household members healthy;Schedule a time to follow up and review how the disclosure went.

Most RR/MDR-TB transmission occurs prior to a diagnosis of RR/MDR-TB being made [17]. Many traditional infection prevention measures, however, are enacted after a person has been diagnosed with RR/MDR-TB. Thus, they reinforce stigma and shame [18]. These include separation from other family members, forced hospitalization, and isolation for work, school, or usual household and social activities. There is a role for infection control measures as part of the approach to RR/MDR-TB prevention, but they should not be the sole focus of TB programs and should be provided in a compassionate manner. Studies have shown that when a person is started on and able to take effective treatment for RR/MDR-TB, there is almost no ongoing transmission even after 48–72 h on medication [19]. Ironically, fear-based infection control strategies might be associated with alienation from the health care system [20], leading to drop out from treatment and an increased risk of TB transmission.

The principles of what has traditionally been called preventive therapy but what should more accurately be referred to as the treatment of infection for drug-susceptible TB and RR/MDR-TB are essentially the same [21]. The goal is identifying individuals when they are harboring only a small population of mycobacteria in the lungs—and thus are usually asymptomatic—so that they can be treated with fewer medications (often just one or two) and for shorter periods of time (ranging from one to six months) than are needed for people with active TB disease. This strategy has been shown to reduce morbidity and mortality for drug-susceptible TB across multiple populations and settings [22,23,24]. Older models of latent TB, which can be prevented from becoming active, are not in keeping with what is now understood about the spectrum of TB infection and disease [25]. In newer models based on improved understandings of pathophysiology, TB is no longer understood to be in binary active or latent states but rather to present in the lungs in insufficient numbers to cause symptoms/disease [26]. The older conceptions of latent or sleeping TB may have contributed to a lack of urgency and resources targeted toward prevention on the part of health systems, policy makers, providers, and programs. 

For both drug-susceptible and RR/MDR *M.tb* infection treatment, the first step is ruling out active TB disease. This is usually conducted through a symptom assessment, weight monitoring, physical examination, bacteriologic testing, and chest radiography [27]. Tests for *M.tb* infection perform poorly as they require a functioning immune system and are, thus, more likely to yield false-negative results in the populations most at risk of becoming sick with TB, such as in young children or persons living with HIV [28]. In addition to better tests of infection, well-quantified exposure scales for TB can serve as proxy measures for TB infection [29]. RR/MDR *M.tb* infection can be presumed if the index individual is sick with a strain of RR/MDR-TB (either confirmed or possible).

What differs between the treatment of infection for drug-susceptible and RR/MDR-TB is the medications that can be used for therapy. By definition, RR/MDR-TB strains are resistant to the medications used to prevent drug-susceptible TB, including rifamycins and (for persons with MDR-TB) isoniazid. For this reason, fluoroquinolones have been the mainstay in the treatment of RR/MDR *M.tb* infection [30]. When the first observational studies of RR/MDR *M.tb* infection treatment were conducted several decades ago, access to drug susceptibility testing for fluoroquinolones was limited and took weeks to months to obtain. This led to the use of multidrug regimens for RR/MDR *M.tb* infection treatment, a strategy that increased both the pill burden and the likelihood of adverse events [31]. In spite of the challenges of multidrug regimens, preventive therapy regimens for RR/MDR-TB were shown to be effective in several small observational studies. Many of these studies were from Cape Town, South Africa, and utilized levofloxacin in combination with high-dose isoniazid and either ethambutol or ethionamide [32]. A large outbreak in the Federated States of Micronesia found fluoroquinolone-based preventive therapy to be both effective and cost-effective and contributed to the World Health Organization making a conditional recommendation for RR/MDR-TB preventive therapy in 2018 [33]. A combination therapy strategy for RR/MDR *M.tb* infection was used in a larger cohort of persons exposed to RR/MDR-TB in the household in Karachi, Pakistan and found the treatment of infection to be effective but that the use of ethionamide resulted in poor adherence, largely mitigated by adverse events [34,35]. Of note, although fluoroquinolones are a mainstay in RR/MDR-TB preventive therapy, other options have been reported in observational cohorts, including high-dose isoniazid [36] or bedaquiline or delamanid given for six months. Short courses (i.e., one month) of linezolid have also been reported on a patient-by-patient basis [37]. Of note, if there is clear evidence that a person has rifampicin mono-resistant *M.tb* infection, isoniaizid-based preventive therapy regimens lasting six months should be effective.

Very few of the persons who could benefit from RR/MDR *M.tb* infection treatment globally have access to it. Currently, RR/MDR *M.tb* infection treatment is usually only implemented in low-burden TB settings. Some of this may be due to a lack of randomized controlled trials leading to a rather lukewarm policy recommendation from the WHO on RR/MDR-TB preventive therapy. However, given the notable morbidity and mortality of RR/MDR-TB disease worldwide and existing programmatic data the balance of risk/benefit ratio would likely favor wider-scale implementation of RR/MDR *M.tb* treatment of infection even while randomized trial data are being collected. There are three ongoing trials of RR/MDR *M.tb* infection treatment, two of which are assessing the role of fluoroquinolones and one of which is testing the novel nitroimidazole medication delamanid [38]. The status of these trials is summarized in Table 1 below. In addition to these trials, there is a need for exploring other models of drug development for RR/MDR *M. tb* infection, including inhaled therapies, vitamin supplementation, and nutritional support.

Much of what is driving the global RR/MDR-TB crisis is poverty, hunger, lack of housing, and co-morbid conditions, such as HIV, substance use disorder, malnutrition, and diabetes [39]. Effective RR/MDR-TB post-exposure management must, therefore, take into account the broad range of issues faced by populations who are at increased risk of the disease and approach them in an integrated and person-centered fashion [40]. While these issues were not reviewed in detail in this paper, they are essential to RR/MDR-TB prevention efforts. Figure 1 summarizes key elements of RR/MDR-TB prevention.

## 5. Khayelitsha Model of Care

All too often, programs and providers report feeling too overwhelmed to take on TB prevention, and this may be even truer for the prevention of RR/MDR-TB. At the same time, RR/MDR-TB prevention activities are critical, since even with improved diagnostic and treatment strategies, outcomes for people living with RR/MDR-TB are far below those achieved for people with drug-susceptible form of TB. While many high-risk TB settings face unique issues and challenges, there are often cross-cutting themes of poverty, injustice, and inequity driving the transmission of RR/MDR-TB in communities. We were able to successfully implement a program aimed at RR/MDR-TB prevention in a peri-urban township outside of Cape Town, South Africa beginning in 2020. With the aim of sharing lessons learnt with policy makers and TB care providers, below is a description of the program as well as the key factors supporting its success [41].

The Khayelitsha post-exposure management program (referred to locally as the “PEP”) was implemented as a collaborative effort between Médecins Sans Frontières, the City of Cape Town, and the Western Cape Department of Health. Household contacts (particularly children and adolescents) of persons newly diagnosed with RR/MDR-TB, were assessed for signs or symptoms of TB, and provided TB counselling, either in their home or at their primary health care clinic. This was conducted by a PEP-dedicated TB nurse who worked across all 10 Khayelitsha clinics. Contacts with any suggestive signs or symptoms of TB were referred for further evaluation by a doctor and, where indicated, for chest radiograph and/or bacteriological testing. Following best practices in the South African setting, tests of infection were not routinely performed given that all these individuals were close contacts of persons newly diagnosed with TB. Contacts without clinical evidence of active TB disease would be offered 6 months of treatment of infection most commonly with a fluoroquinolone (high-dose isoniazid or delamanid was used where there was suspected or confirmed fluoroquinolone resistance in the index case).

Patient-centered and differentiated care was a key aspect of this program (this was particularly essential given that this program was rolled out at the same time as the COVID pandemic hit), with persons given the option to follow up for treatment of infection in their home, at the clinic, or telephonically. Follow-up consults focused on ongoing treatment literacy/counseling support, and eliciting if there were any emerging TB symptoms or adverse events. To minimize disruptions to people’s lives, follow-up visits were conducted every two to three months, with multi-month medication refills provided. Young children were offered child friendly formulations of medication to support treatment adherence, including dispersible tablets of levofloxacin provided by the Global Drug Facility. Where socio-economic concerns were noted in the household, persons could be referred to a TB counsellor or social worker and could be provided with food parcels. Transport allowance was provided where needed for travel to attend any clinic visits.

The outcomes of the initial phase of this program (112 children and adolescents on treatment of infection) are described in more detail in a recent publication. Notably, this program resulted in a high level of case detection (10% of those screened were started on treatment for TB disease), as well as high retention in care for those who received treatment for infection (80% completed treatment); therapy was well tolerated with minimal adverse events. None of those who started treatment for infection developed active TB disease (median 200-day follow-up). After the initial pilot success of this program, the work was integrated into the routine TB services offered at the clinic.

## 6. Conclusions

It is imperative that more work is conducted to prevent RR/MDR-TB in the coming years. RR/MDR-TB is one of the most serious aspects of the global crisis in antimicrobial resistance, but efforts to prevent RR/MDR-TB have narrowly focused on resistance amplification, while ignoring other important interventions. RR/MDR-TB preventative efforts require a fundamental shift in the thinking and actions of the TB community, to a focus on providing person and family-centered care along the whole continuum. Rapid diagnosis of all forms of TB and initiation of the shortest most effective regimen is key to RR/MDR-TB prevention. So too is compassionate post-exposure care for all close contacts and members of the family coupled with supportive counselling. In this review, we have outlined key preventative strategies along the spectrum of care and shared our experiences of a family centered RR/MDR-TB preventive care model. Implementation of these strategies by committed TB programs and care providers promises to dramatically improve RR/MDR-TB prevention efforts and could, with time, reduce the implementation burdens of treating RR/MDR-TB for countries and programs. Doing so will dramatically reduce the suffering of persons, such as Busisiwe and her family.

## Figures and Tables

**Figure 1 pathogens-12-00728-f001:**
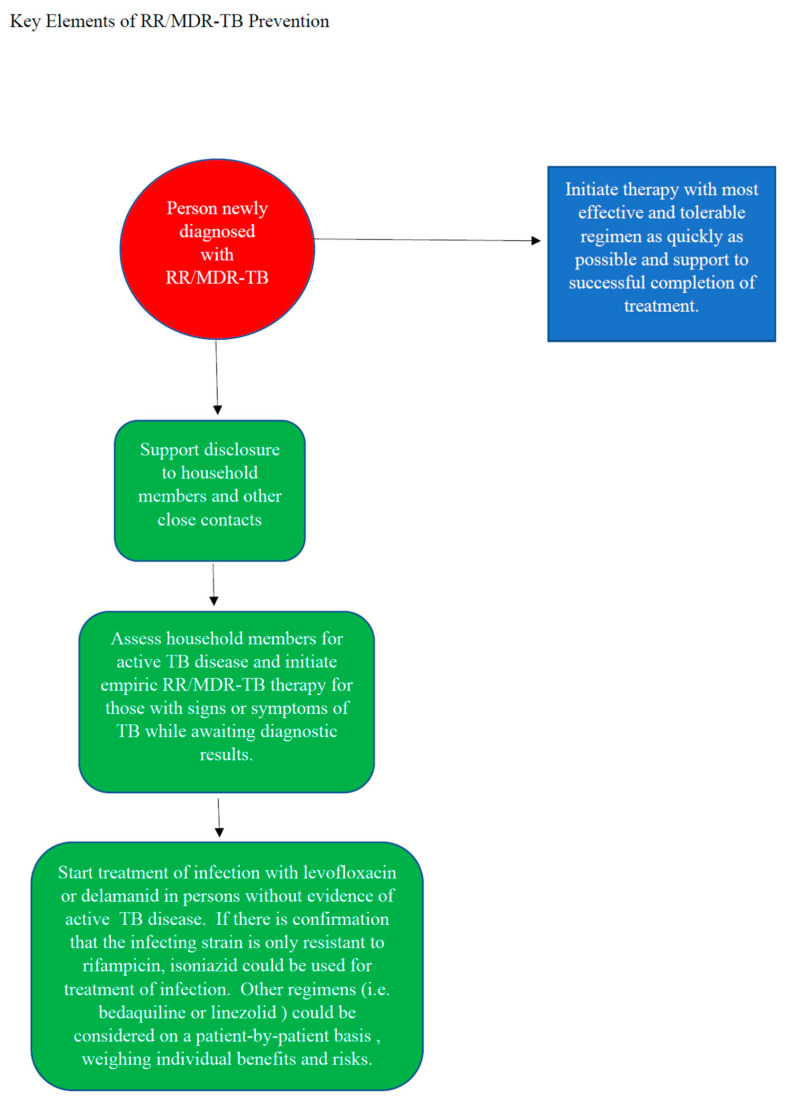
A summary of the key elements of RR/MDR-TB prevention.

**Table 1 pathogens-12-00728-t001:** Ongoing randomized Controlled Trials for RR/MDR *M.tb* Treatment of Infection.

Study Name	Registration Number	Eligible Populations	Regimens Being Assessed	Comments
V-QUIN	ACTRN12616000215426	Household contacts of all ages of persons diagnosed with RR/MDR-TB	Levofloxacin versus placebo	Enrollment is completed and results are likely to be analyzed as pooled data with TB-CHAMP, results expected in 2025
TB-CHAMP	ISRCTN92634082	Household contacts ages 18 years and under of persons diagnosed with RR/MDR-TB in South Africa	Levofloxacin versus placebo	Enrollment is completed and results are likely to be analyzed in pooled data with V-QUIN, results expected in 2025
PHOENIx MDRTB	NCT03568383	Household contacts of persons newly diagnosed with RR/MDR-TB, multicentered trial	Delamanid versus isoniazid	Enrollment is ongoing and results are likely expected in 2026

## Data Availability

Not applicable.

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
