# Peer review of "Treatment of Infection as a Core Strategy to Prevent Rifampicin-Resistant/Multidrug-Resistant Tuberculosis"

_pathogens, 2023, doi:10.3390/pathogens12050728_

Round 1

Reviewer 1 Report (Previous Reviewer 2)

I would like to thank the authors for providing thorough responses to all my questions and concerns, the manuscript looks much better in the revised form.

As a minor comment, I would recommend addressing the infection with RR/MDR M. tuberculosis strains, but not yet the disease, as "RR/MDR M. tuberculosis infection", or "RR/MDR Mtb infection", rather than "RR/MDR-TB infection", as TB implies the tuberculosis disease. Perhaps you could point this difference in the introduction, and section 3.

Also, after the renumbering of references, reference #7 is missing. Reference #8 seems to be a very local case, perhaps the authors might consider adding a couple more references about the compensatory mutations, or citing a review on this issue.

Author Response

  1. As a minor comment, I would recommend addressing the infection with RR/MDR M. tuberculosis strains, but not yet the disease, as "RR/MDR  tuberculosis infection", or "RR/MDR Mtb infection", rather than "RR/MDR-TB infection", as TB implies the tuberculosis disease. Perhaps you could point this difference in the introduction, and section 3.

Thank you.  We have made this change throughout the paper.  We have also introduced this concept in section 3.

  1. Also, after the renumbering of references, reference #7 is missing. Reference #8 seems to be a very local case, perhaps the authors might consider adding a couple more references about the compensatory mutations, or citing a review on this issue.

We have reviewed the references and number 7 is there.  We have also changed this reference to the following:

Gagneux, S., Davis Long, C., Small, P., et al. The competitive cost of antibiotic resistance in Mycobacterium tuberculosis.  Science 2006, 312(5782): 1944-6.

Reviewer 2 Report (Previous Reviewer 1)

All my previous comments have been address appropriately

Author Response

Thank you for reviewing the revised paper.

Reviewer 3 Report (New Reviewer)

The incidence of drug resistant tuberculosis is a serious threat to global health and the focus of significant drug discovery efforts. Cures have not been identified in over 100 years of research and treatments are elusive. The latter take decades to emerge and only then as less than optimal combinations of high doses of drugs. Alternative prevention and palliative strategies could improve the quality of life for individuals and communities relatively quickly and create a breadth of care that does not leave a gap while treatments are under development.

This manuscript is a timely reminder to researchers of the human cost of the absence of adequate treatment and control of disease. The effect on families and local communities is devastating and the need for a solution is near and present, invoking a level of urgency that may not be adequately appreciated even by those conducting research and developing products. Consequently, this manuscript raises awareness in a manner that is valuable to the field.

It is not clear if it was the intent of the authors or the journal to include the corrections to the manuscript which made it a little difficult in places to read.

My recollection of early strain genotyping studies conducted in a township outside Cape Town was the surprising finding that a single strain might be in multiple households at geographically disparate locations within the community and that a single household might house individuals, each having different strains. This, for me, was counter intuitive as one would ordinarily think of a center from which the disease spreads and by inference ease of control once an infected individual is identified. Would this point be of value to the authors in strengthening their case for a need for improved management strategies?

P3, last para – Would lung therapy help in treating the disease before it is established? A wide variety of inhaled therapies are in various stages of development, but the approach generally meets resistance from the TB community due to the systemic nature of disease, despite its use in the mid-20th Century. I raise this point not to persuade the authors of the validity of this therapeutic strategy but to inquire if it is another potentially viable approach to help them address the short-term needs of the individual and the community.

P6 - Could it be argued that the benefits of a concerted effort in preventative care might alleviate the overwhelming nature of implementation for programs and providers if sufficient time was allowed for them to emerge? Is there evidence for these benefits that could be extrapolated to support this observation?

Author Response

Mycobacterium tuberculosis.  Science 2006, 312(5782): 1944-6.

  1. My recollection of early strain genotyping studies conducted in a township outside Cape Town was the surprising finding that a single strain might be in multiple households at geographically disparate locations within the community and that a single household might house individuals, each having different strains. This, for me, was counter intuitive as one would ordinarily think of a center from which the disease spreads and by inference ease of control once an infected individual is identified. Would this point be of value to the authors in strengthening their case for a need for improved management strategies?

Thank you for this comment.  To address this, we have added the following sentence to the second paragraph in section 4: “Contact tracing could also be expanded beyond the household and into entire communities with high rates of RR/MDR-TB using the same strategies we describe here.”

  1. P3, last para – Would lung therapy help in treating the disease before it is established? A wide variety of inhaled therapies are in various stages of development, but the approach generally meets resistance from the TB community due to the systemic nature of disease, despite its use in the mid-20th I raise this point not to persuade the authors of the validity of this therapeutic strategy but to inquire if it is another potentially viable approach to help them address the short-term needs of the individual and the community.

Thank you for this comment.  To address it, we have added the following sentence on page 4: .   In addition to these trials, there is a need for exploring other models of drug development for RR/MDR M. tb infection, including inhaled therapies, vitamin supplementation, and nutritional support.

  1. P6 - Could it be argued that the benefits of a concerted effort in preventative care might alleviate the overwhelming nature of implementation for programs and providers if sufficient time was allowed for them to emerge? Is there evidence for these benefits that could be extrapolated to support this observation?

Thank you.  To address this comment, we have added the following sentence to the conclusion:

“Implementation of these strategies by committed TB programs and care providers promises to dramatically improve RR/MDR-TB prevention efforts and could, with time, reduce the implementation burdens of treating RR/MDR-TB for countries and programs.”

This manuscript is a resubmission of an earlier submission. The following is a list of the peer review reports and author responses from that submission.

Round 1

Reviewer 1 Report

This is a very well written perspective. I only have few minor comments

1- In the box page 3. "Assess any physician or mental risks ..." should read "Assess any physical or mental risks ..."

2- Conclusion. page 7. "the shortest most effective and regimen" should read "the shortest most effective regimen"

3- Reference. please check the references and their indexing. e.g. the indexing go through 43. but their are only 40 citations listed. 

Very good

Author Response

Responses to Reviewers

Reviewer 1

  • In the box page 3. "Assess any physician or mental risks ..." should read "Assess any physical or mental risks ..."

Thank you.  We have corrected this.

  • page 7. "the shortest most effective and regimen" should read "the shortest most effective regimen"

Thank you, we have corrected this.

3- Reference. please check the references and their indexing. e.g. the indexing go through 43. but their are only 40 citations listed. 

Thank you for this comment.  We have updated the references.

Reviewer 2

  1. The article presented by A. Reuter and J. Furin is submitted as a review article, thus it is supposed to summarize and review some data on a specific subject, namely the strategies to treat and prevent the spread of RR/MDR-TB. However the paper reviews only a couple of clinical trials on preventive therapy for RR/MDR-TB in South Africa, and represents an extended essay to promote the strategy proposed by the authors. The paragraph about Busisiwe’s family (lines 31-51) is more relevant to a local newspaper, while its value to a scientific review is questionable, as the ethical aspect of mentioning a life story of a patient.

Thank you for your comments and your careful review of the paper.  There is a long history of using patient experiences to illustrate examples in medicine, especially when there is only limited data or to illustrate certain points.  This manuscript is meant to be a perspectives piece, and in our experience, presenting this story of an individual who has lived through this experience is essential to humanizing the scientific points raised in the paper.  We therefore feel it is important to keep the personal story in.  it is unclear to us why this would be considered “unethical”.  We have the permission of the person to use the story and we have added this comment in as a footnote.   We have followed standard best practices for publishing case reports in terms of privacy, consent, and authorship and therefore we feel we have met ethical standards (see the following paper at: https://www.ncbi.nlm.nih.gov/pmc/articles/PMC8493572/).

  1. Some parts of the paper are misleading, such as the statement that “current tests for TB infection cannotdifferentiate between drug-susceptible and drug-resistant strains” on lines 53-56. The Xpert MTB/RIF assay may be used both to detect Mtb and detect Rif resistance. Other tests are also available on market, which differentiate drug-susceptible and drug-resistant Mtb strains.

Thank you for this comment.  We are aware of these tests, but they are only able to differentiate between drug-susceptible and drug-resistant TB DISEASE.  This is because they require the actual isolation of bacterial or bacterial DNA (in the case of molecular tests).  But this paper is about people who do not have TB disease but rather who have TB infection.  TB infection is characterized by an inability to isolate the bacteria.  And because the bacteria cannot be isolated, drug susceptibility testing cannot be done.  So while these tests do exist for people with TB disease, they cannot be used for TB infection. To clarify tis, we have added the following sentence: “This is because drug susceptibility testing can only be done when M. tuberculosis or its DNA has been isolated.”

  1. The authors cannot make a point what they are calling “preventive therapy”, and what - “treatment of infection”, constantly putting these terms in quotes (lines 127-129), which is also more of a journalistic style, rather than scientific. The authors question the definitions of “latent” TB (lines 134-138), however don’t provide evidence of any better theories. 

We have added this sentence:

“In newer models based on improve understandings of pathophysiology, TB is no longer understood to be in binary active or latent states but rather to present in the lungs in insufficient numbers to cause symptoms/disease.”

We have added reference to what is now widely considered the prevailing spectrum of disease model for TB (see https://www.ncbi.nlm.nih.gov/pmc/articles/PMC6217958/).  We have removed the quotes.

  1. The authors also state, that there is a “legacy” passed down by “paternalistic and neo-colonial public health experts” that DR strains are less virulent and transmissible, in a “total absence of data”. I would suggest the authors to read some papers on the fitness-cost of DR-mutations, and on compensatory mutations, before claiming this a “paternalistic erroneous legacy”. To prove their point of view, the authors cite their own paper, dated back to 1999 (reference #7).

We have edited this paragraph to remove this sentence and added a citation looking at fitness costs..

  1. The issue of self-citing is also very grave in this paper. 7 out of 40 references (almost 20%) are self-citations (references 7, 15, 16, 20, 21, 27, 40). This seems to be very unethical. Not to say that 40 references is very little for a review.

We thank the reviewer for pointing this out.  To address this reviewer’s concerns, we have either removed the citations or tried to find alternatives.  We had to leave one of the citations of our own work in as it was using data from a paper we published (reference 41).

  1. Overall, this review does not provide any contribution to the field, nor does it summarize any objective data. It’s style is more journalistic, rather then scientific, some sentences are very long and hard to understand, some points are not clear. The ethical aspect of this paper in terms of self-citing is also very questionable. 

We apologize that the reviewer thinks the paper does not add anything to the field.  As a perspectives piece, we felt we had more room with the style of the paper. We feel we have summarized the data that is out there and synthesized the literature from DS-TB and DR-TB (see lines 165-79).  We have also included ongoing trials (see Table 1) since there are not yet any trial data to review.

  1. Some sentences are very long and hard to understand, some grammatical errors and typos are also present.

We have edited the paper for grammar and style.

Reviewer 2 Report

The article presented by A. Reuter and J. Furin is submitted as a review article, thus it is supposed to summarize and review some data on a specific subject, namely the strategies to treat and prevent the spread of RR/MDR-TB. However the paper reviews only a couple of clinical trials on preventive therapy for RR/MDR-TB in South Africa, and represents an extended essay to promote the strategy proposed by the authors. The paragraph about Busisiwe’s family (lines 31-51) is more relevant to a local newspaper, while its value to a scientific review is questionable, as the ethical aspect of mentioning a life story of a patient.

Some parts of the paper are misleading, such as the statement that “current tests for TB infection cannot differentiate between drug-susceptible and drug-resistant strains” on lines 53-56. The Xpert MTB/RIF assay may be used both to detect Mtb and detect Rif resistance. Other tests are also available on market, which differentiate drug-susceptible and drug-resistant Mtb strains.

The authors cannot make a point what they are calling “preventive therapy”, and what - “treatment of infection”, constantly putting these terms in quotes (lines 127-129), which is also more of a journalistic style, rather than scientific. The authors question the definitions of “latent” TB (lines 134-138), however don’t provide evidence of any better theories. 

The authors also state, that there is a “legacy” passed down by “paternalistic and neo-colonial public health experts” that DR strains are less virulent and transmissible, in a “total absence of data”. I would suggest the authors to read some papers on the fitness-cost of DR-mutations, and on compensatory mutations, before claiming this a “paternalistic erroneous legacy”. To prove their point of view, the authors cite their own paper, dated back to 1999 (reference #7).

The issue of self-citing is also very grave in this paper. 7 out of 40 references (almost 20%) are self-citations (references 7, 15, 16, 20, 21, 27, 40). This seems to be very unethical. Not to say that 40 references is very little for a review.

Overall, this review does not provide any contribution to the field, nor does it summarize any objective data. It’s style is more journalistic, rather then scientific, some sentences are very long and hard to understand, some points are not clear. The ethical aspect of this paper in terms of self-citing is also very questionable. 

Some sentences are very long and hard to understand, some grammatical errors and typos are also present.

Author Response

(The authors gave the same response as above.)
